# LINA: An LLM-driven Neuro-Symbolic Approach for Faithful Logical Reasoning

## Abstract

Large Language Models (LLMs) have exhibited remarkable potential across a wide array of reasoning tasks, including logical reasoning. Although massive efforts have been made to empower the logical reasoning ability of LLMs via external logical symbolic solvers, crucial challenges of the poor generalization ability to questions with different features and inevitable question information loss of symbolic solver-driven approaches remain unresolved. To mitigate these issues, we introduce **LINA**, a LLM-driven neuro-symbolic approach for faithful logical reasoning. By enabling an LLM to autonomously perform the transition from propositional logic extraction to sophisticated logical reasoning, LINA not only bolsters the resilience of the reasoning process but also eliminates the dependency on external solvers. Additionally, through its adoption of a hypothetical-deductive reasoning paradigm, LINA effectively circumvents the expansive search space challenge that plagues traditional forward reasoning methods. Empirical evaluations demonstrate that LINA substantially outperforms both established propositional logic frameworks and conventional prompting techniques across a spectrum of five logical reasoning tasks. Specifically, LINA achieves an improvement of 24.34% over LINC on the FOLIO dataset, while also surpassing prompting strategies like CoT and CoT-SC by up to 24.02%. Our code is available at `https://anonymous.4open.science/r/nshy-4148/`.

## 1 Introduction

Large language models (LLMs) have exhibited remarkable capabilities across a wide range of NLP tasks (Achiam et al., 2023; Anil et al., 2023; Touvron et al., 2023), sometimes even outperforming human levels of performance. Nevertheless, these advanced models struggle with mathematical and complex logical reasoning tasks (Arkoudas, 2023; Liu et al., 2023). The Chain-of-Thought (CoT) prompting technique (Kojima et al., 2022; Wei et al., 2024; Nye et al., 2021) has emerged as an effective strategy to enhance reasoning skills by incorporating intermediate steps into the reasoning process. Building on this foundation, subsequent studies have developed methodologies like LAMBADA (Kazemi et al., 2023), Tree-of-Thought (ToT) (Yao et al., 2024), and Chain-of-Thought with Self-Consistency (CoT-SC) (Wang et al., 2023). Despite these advancements, recent studies (Bao et al., 2024a; Lanham et al., 2023; Lyu et al., 2023; Turpin et al., 2024) highlight that LLMs continue to face challenges in maintaining faithful reasoning processes, where even logically sound chains do not guarantee accurate outcomes. To address unfaithful reasoning in complex tasks, methods like Faithful Chain-of-Thought (Lyu et al., 2023), LINC (Olausson et al., 2023), Logic-LM (Pan et al., 2023), and SatLM (Ye et al.) have been proposed. These approaches translate logical problems into formal expressions and use external symbolic solvers to produce symbolic results, which are subsequently interpreted by large language models (LLMs) or dedicated interpreters.

While these neuro-symbolic techniques effectively mitigate unfaithful reasoning, they also present several challenges. First, the process of converting logical problems into formal expressions leads to **information loss**. This information loss may stem from certain contextual information or from information that cannot be effectively converted due to the limited expressive power of the chosen formal representation. For example, in a neuro-symbolic approach that combines first-order logic (FOL) and FOL solvers, important information behind predicate definitions can be lost during the line-by-line conversion of a logical problem into FOL. Consider the problem: "A is east of B, C is west of B, determine if A is east of C". When we use the definitions "E(x, y): x is east of y; W(x, y):

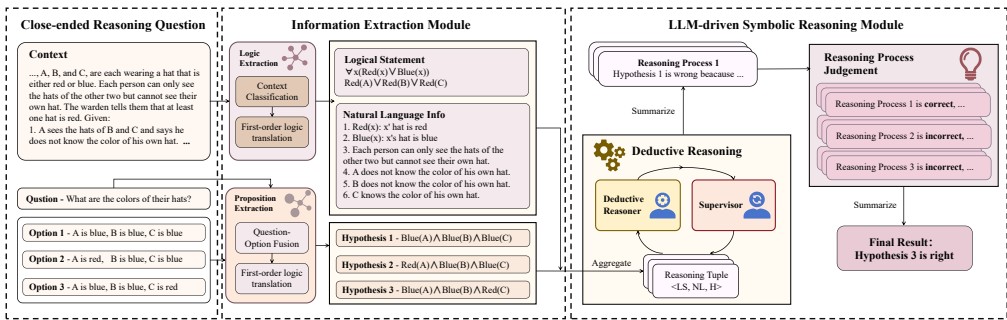

Figure 1: The framework of the **LLM-driven Neuro-Symbolic Approach for Faithful Logical Reasoning** consists of two main components: the Information Extraction Module and the LLM-driven Symbolic Reasoning Module. The close-ended reasoning questionon the left is processed by the Information Extraction Module to generate first-order logic statements ($LS$), natural language information ($NL$), and hypothesis ($H$). These outputs are then fed into the LLM-driven Symbolic Reasoning Module on the right, which performs deductive reasoning to derive the final answer.

x is west of y" to convert the problem into first-order logic expressions, we get $E(A, B) \wedge W(C, B)$, and we need to prove $E(A, C)$. Directly inputting these FOL expressions into the solver will return the result *Uncertain*, because the conversion process loses the logical information embedded in the predicates $E$ and $W$, such as $W(C, B) \rightarrow E(B, C)$ and $E(A, B) \wedge E(B, C) \rightarrow E(A, C)$. This loss of logical information is critical for solvers that strictly rely on the input. Even humans and LLMs with background knowledge need to be explicitly informed of the definitions of $E(x, y)$ and $W(x, y)$ in order to correctly answer the translated question. Second, the reliance on specific external tools results in **poor generalization** of these methods, limiting them to solving only certain types of problems, such as FOL propositional inference problems (Olausson et al., 2023) or satisfiability problems (Ye et al.).

To address these challenges, we propose an LLM-driven neuro-symbolic approach for faithful logical reasoning, **LINA**, as shown in Figure 1. This method leverages a carefully designed information extraction strategy to mitigate the issue of information loss. Additionally, LINA integrates a meticulously crafted LLM-based deductive reasoning algorithm, eliminating the dependency on external tools while reducing the unfaithful risks associated with purely LLM-based reasoning. Specifically, the architecture of LINA comprises two core components: the *Information Extraction* module and the *LLM-driven Neuro-Symbolic Reasoning* module. In the information extraction module, LINA addresses the challenge of information loss by incorporating first-order logic (FOL), commonly used in existing neuro-symbolic paradigms, while preserving important natural language information that cannot be directly captured by FOL's logical expressions. This strategy not only aids the subsequent reasoning module in delivering more reliable reasoning based on FOL's rich inference rules, but also ensures that the module effectively captures all valid information from the logical problem.In the LLM-driven neuro-symbolic reasoning module, LINA tackles the generalization issues caused by the reliance on external tools by introducing a deductive reasoning method that relies solely on LLMs, eliminating the need for external resources. By reformulating closed-form logical reasoning tasks as deductive reasoning challenges, it guides the LLM through a structured, step-by-step reasoning process based on FOL rules, utilizing background information and hypotheses until contradictions are identified or consistency is established. This deductive reasoning approach, which integrates both FOL and natural language information, enhances the reliability of the LLM-only reasoning process by introducing inference rules and task reformulation. Throughout the process, LINA employs multiple verification mechanisms to further enhance the trustworthiness of the reasoning outcomes.

To validate the effectiveness of LINA, we conduct various evaluations across five datasets. We compare the performance of LINA with existing neuro-symbolic methods such as SatLM and LINC, particularly on more diverse and complex datasets like LogiQA, where our method achieve performance improvements of up to 35.24% over these approaches. This comparison demonstrates the significant advantage of LINA in terms of generalization. Through a case study, we demonstrate that LINA effectively resolves the issue of information loss during the information extraction process

in neuro-symbolic methods. Additionally, we compare the performance of LINA with prompting methods such as CoT, CoT-SC, and ToT, achieving accuracy improvements of up to 24.34%, indicating that LINA is more faithful than these approaches. An ablation study further validate the effectiveness of several key strategic choices in LINA.

The contributions of this paper are as follows:

1. We propose an innovative neuro-symbolic method, LINA, which uses hypothetical-deductive reasoning and leverages LLMs for symbolic inference, which addresses the issues of deployment complexity and information loss in existing symbolic methods.

2. We provide the graph interpretation of LINA. Based on this, we also provide the theoretical property and complexity analyses of LINA.

3. We conducted extensive experiments to evaluate the effectiveness of the LINA method, demonstrating its superiority over existing neuro-symbolic and prompting-based methods.

## 2 RELATED WORK

**Prompt-based LLM Reasoning.** Logical reasoning, which aims to draw truthful conclusions from given condition, premises and contexts, is a fundamental task for the application of LLMs (Mondorf & Plank, 2024; Sun et al., 2023; Qiao et al., 2023). Prompting is a direct and effective technique for stimulating the logical reasoning ability of LLMs. Considering the complexity of reasoning questions, one significant direction of prompt-based reasoning is step-by-step reasoning (Besta et al., 2024b; Wang et al., 2023). Wei et al. (2024) proposed the Chain-of-Thought (CoT) technique that enables LLMs to output the reasoning process step-by-step. Along this line, Yao et al. (2024) proposed the Tree-of-Thought (ToT) technique. It enables LLMs to self-evaluate and choose between various reasoning paths, therefore empowering their reasoning ability. Besta et al. (2024a) proposed the Grapth-of-Thought (GoT) method that further improves the reasoning performance of LLMs on complex questions. However, these methods fall short in non-mathematical reasoning (Sprague et al., 2024) and cases where the complexity of exemplars and target questions differs a lot. Another pivotal direction in prompt-based LLM reasoning is the question decomposition (Zhang et al., 2023; Yao et al., 2023; Kazemi et al., 2023). Zhou et al. (2023) proposed the least-to-most prompting strategy, which breaks down complex problems into series of simpler sub-problems and solves them in sequences. Cui et al. (2023) proposed the divide-and-conquer-reasoning approach for consistency evaluation of LLMs. Zhang et al. (2024) then proposed to apply the divide-and-conquer strategy to enhance the logical reasoning ability of LLMs. To address the challenge of huge search spaces of step-by-step reasoning, Kazemi et al. (2023) proposed a backward chaining algorithm that decomposes reasoning into sub-modules that are easier for LLMs to solve. Despite their success in enhancing the reasoning ability of LLMs in specific tasks, existing prompting-based algorithms confront with problems of expensive costs and unstable reasoning performance (Yao et al., 2024).

**Symbolic Methods for Logical Reasoning.** Symbolic logical reasoning techniques utilize symbolic logical symbols and expressions for consistent and accurate reasoning, which overcomes the inconsistent reasoning and ordering sensitivity challenges of prompt-based reasoning algorithms (Chen et al., 2024; Bao et al., 2024a). One key idea is to enhance the reasoning ability of LLMs via invoking logical symbols and expressions (Wang et al., 2022; Wan et al., 2024). Along this line, Wang et al. (2022) proposed a symbol-enhanced text reasoning framework that extends natural language problems to logical symbols and expressions to enhance logical answer matching. Bao et al. (2024b) proposed a logic-driven data augmentation approach that transforms problem texts to structured semantic graphs to enhance language model-based reasoning frameworks. Another pivotal idea is to transform textualized problems into logical expressions via LLMs, then solve them with symbolic logic solvers (Olausson et al., 2023; Pan et al., 2023; Ye et al.). The choice of logic solvers, such as SAT solver (Ye et al.) or first-order logic solver (Pan et al., 2023), highly affects the accuracy and generalization ability of reasoning algorithms given datasets with different features. Despite their brilliant performance in consistent logical reasoning, symbolic methods usually confront with information loss in logical expression extraction, which can lead to inevitable reasoning ability dropping (Pan et al., 2023).

## 3 PRELIMINARY

**Task Definition**. This study focuses on the close-ended reasoning task, which is common in real-world application scenarios of LLMs. Specifically, let each reasoning question consists of a context $C$, a question text $Q$, and an option set $O = \{o_1, o_2, \ldots, o_n\}$. The goal of the close-ended reasoning task is to extract a subset $O'$ from the option set, such that for each option $o \in O'$, it is logically non-contradictory with context $C$ and question $Q$.

## 4 METHODOLOGY

### 4.1 OVERVIEW

The structure of LINA is shown in Figure 1. The key idea of this approach is to first transform the textual information of the logical reasoning problem, and then apply a hypothetical-deductive method based on LLMs combined with first-order logic rules. This addresses key challenges such as information loss and deployment difficulties in previous neuro-symbolic methods, as well as the unfaithful reasoning seen in LLMs prompting. Specifically, it consists of the Information Extraction module and the LLM-driven Symbolic Reasoning Module, as illustrated in the Information part and LLM-driven Symbolic Reasoning part of Figure 1, respectively.

### 4.2 INFORMATION EXTRACTION MODULE

The information extraction module takes the logical problem's context, question, and options as input, performs key information extraction and transformation, and outputs a Reasoning Tuple $< LS, NL, H >$, which represents for logical statements, natural language information and hypotheses.

The critical design issue of this module is determining how to represent the extracted information in a way that allows it to be effectively utilized by the subsequent reasoning module, while minimizing the risk of unfaithful reasoning. Additionally, the module should avoid information loss during the transformation process.

This module integrates first-order logic (FOL) and natural language extraction. Converting logical statements into FOL enables the subsequent reasoning process to leverage FOL's rich rules for logical inference, rather than relying solely on semantic reasoning in natural language. This rule-based, formalized approach makes the reasoning process more reliable and easier to verify. The condensed natural language information ensures that the semantic integrity of the text is maintained, preserving elements of the problem that may not easily be expressed in FOL. This strategy effectively addresses the issue of limited expressive power caused by solely using first-order logic expressions, while also preserving sufficient contextual information for the subsequent reasoning module, thereby avoiding information loss during extraction.

Specifically, The text of the logical reasoning problem stem $Context$ undergoes context classification, where lengthy texts are condensed into shorter sentences. These sentences are then categorized based on their ease of translation into first-order logic (FOL). Following classification, the logical statements are translated into FOL, resulting in Logical Statements, $LS = [ls_{1 \ldots i}]$, comprising FOL statements. Predicate definitions produced during the FOL translation, along with the natural language content retained during classification, form Natural Language Information $NL$.

Additionally, to facilitate the subsequent deductive reasoning process, the semantics of the question and options are integrated into a declarative hypothesis proposition. This hypothesis proposition is then subject to FOL translation, ultimately forming formalized hypothesis statements $H_1, H_2, H_3, \ldots$. At this point, the information extraction module has produced all the necessary elements for the Reasoning Tuple $< LS, NL, H >$.

### 4.3 LLM-DRIVEN SYMBOLIC REASONING MODULE

The objective of the LLM-driven Symbolic Reasoning Module is to reliably solve logical reasoning tasks and produce the correct answer.

This module takes the extracted information as input, performs deductive reasoning, and outputs the final answer. The main challenge is ensuring the reliability of the LLMs during reasoning and designing methods to improve its performance.

To address this, the module employs the hypothetical-deductive method, breaking down closed-choice questions into tasks that prove or refute hypotheses. The reasoning process uses FOL rules and simplify complex problems into manageable steps, enhancing the model's reasoning capabilities. A supervisor monitors the reasoning steps to ensure accuracy, and a Reasoning Process Judgement is used to validate the final answer.

After receiving the Reasoning Tuple $< LS, NL, H >$ from the Info Extraction module, the LLM agent carries out step-by-step deductive reasoning using hypothesis $H$. It identifies relevant information from $LS$ and $NL$ to derive a reasoning result $C_0$ based on FOL rules.

The supervisor checks for errors in the reasoning process and may adjust $C$ or reset $C = H$. It then decides whether to continue reasoning, based on whether $C$ conflicts with $< LS, NL, H >$ (refuting $H$) or if $C$ is already supported by $LS$ or $NL$ (proving $H$). If the process continues, the hypothesis is updated to $H' = C$, and reasoning proceeds until the supervisor reaches a conclusion or the step limit $k$ is met.

We summarize the pseudo-code of the proposed algorithm in Algorithm 1.

---

**Algorithm 1** Deductive Reasoning Process

---

**Require:** Hypothesis $H$, Logic Statements $LS$, Natural Language Information $NL$
**Ensure:** Validity of Hypothesis $H$
 1: **while** not reached step limit $k$ **do**
 2:     $C \leftarrow \text{Deductive}(LS, NL, H)$
 3:     $C \leftarrow \text{Check}(C)$
 4:     **if** $C$ contradicts $LS$, $NL$ or $H$ **then**
 5:         Disprove hypothesis $H$
 6:         **exit**
 7:     **else if** $C$ confirms $LS$, $NL$ or $H$ **then**
 8:         Hypothesis $H$ is validated
 9:         **exit**
10:     **else**
11:         Update hypothesis $H \leftarrow C$
12:     **end if**
13: **end while**

---

If multiple hypotheses are deemed correct, the Final Evaluation examines each reasoning process for logical consistency and FOL rule correctness. The final conclusion is chosen through a majority voting mechanism to ensure reliability.

### 4.4 THE GRAPH INTERPRETATION AND COMPLEXITY ANALYSIS OF LINA

**The Graph Interpretation of LINA**. The reasoning process of the LINA algorithm can be reinterpreted as a form of graph search, which facilitates property analysis. The main idea is that *any closed-form logical reasoning problem can be transformed into a path search problem on a finite graph*. Specifically, given a closed-form logical reasoning problem, we define its graph representation $G = (V, E_c, E_t, E_n)$, where $V$ is the set of all propositions related to the problem; $E_c$ is the set of **black undirected edges** representing logical equivalence relations; $E_t$ is the set of **black directed edges** representing logical entailment relations; and $E_n$ is the set of **red undirected edges** representing logical negation relations. Then, we present the following lemma (proofs are available in the appendix):

**Lemma 1.** *A closed-form logical reasoning task is equivalent to the following task: given a logic graph $G = (V, E_c, E_t, E_n)$, an initial point $s \in V$, and a terminal point $t \in V$, find a path from $s$ to $t$ consisting only of black edges.*

Thus, we can analyze the properties of the graph to obtain the properties of LINA. First, since logical equivalence relations are transitive, each set of equivalent propositions forms a **clique**, which can be

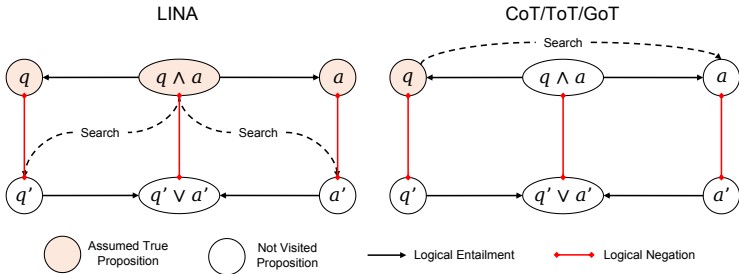

Figure 2: An illustration of the graph interpretation of LINA and prompt-based reasoning. Here $q$ denotes the question proposition, while $a$ denotes the option proposition. The $q'$ denotes $\neg q$, and the $a'$ denotes $\neg a$. The goal of LINA is to find a path from $q \wedge a$ to $q'$ or $a'$ in order to **falsify** $a$. The goal of prompt-based reasoning is to find a path from $q$ to $a$ in order to **verify** $a$.

contracted into a single node. Therefore, a logical reasoning problem can be abstracted into a graph $G = (V_c, E_t, E_n)$ consisting of entailment and negation edges. Here $V_c$ denotes all nonequivalent propositions related to the problem. Next, we present the following lemma:

**Lemma 2.** *The reasoning module of LINA is equivalent to, given a problem proposition $q$ and option proposition $a$, finding a path from $q \wedge a$ to $\neg q$ or $\neg a$ within a finite number of steps.*

Lemma 2 provides a graph-based interpretation of the LINA algorithm, showing that LINA essentially performs a finite-step search on $G$. An example of the graph interpretation is shown in Figure 2. We need to ensure the theoretical correctness of LINA. Therefore, we give the following theorem.

**Theorem 1.** *Given a logical reasoning graph $G = (V_c, E_t, E_n)$, an assumed true proposition $s \in V_c$ and an unvisited proposition $t \in V_c$, if there exists a black directed path from $s$ to $t$, then there cannot exist a path starting from $s \wedge t$, ending at $\neg s$ or $\neg t$.*

Theorem 1 essentially clarifies the validity of the LINA algorithm. If we replace $s$ with $q$ and replace $a$ with $t$ in Theorem 1, we immediately conclude that if the option proposition $a$ is entailed by the question proposition $s$, then there does not exist any path from $q \wedge a$ to $\neg q$ or $\neg a$, which is the search goal of LINA. In other words, **LINA theoretically can identify false options in finite steps.**

**Complexity Analysis**. The complexity of LINA could be deduced using its graph interpreation. For LINA, assuming the number of search steps $S > |E_t|$, the graph search process traverses the black directed edges $E_t$ without revisiting nodes; additionally, red directed edges appear at most once in the search path. Therefore, given the search graph $G = (V, E_t, E_n)$, the time complexity of the LINA algorithm is $O(|E_t|)$, which is comparable to the time complexity of algorithms like ToT (Yao et al., 2024). Moreover, as shown in Figure 2, LINA has more target vertices ($q'$ and $a'$) and more assumed true (visited) propositions ($q$, $a$ and $q \wedge a$) than prompt-based algorithms. **Merely finding one path from the source to one target is enough for finishing the graph search.** Therefore, it is easier for LINA to finish the graph search than prompt-based algorithms.

## 5 EXPERIMENT

### 5.1 EXPERIMENTAL SETUP

**Datasets.** In our experiments, we selected five datasets commonly used in logical reasoning research: (1) **ReClor** (Yu et al., 2020): ReClor is a reading comprehension dataset requiring logical reasoning, composed of logical reasoning and related problems extracted from standardized examinations such as the Law School Admission Test (LSAT) and the Graduate Management Admission Test (GMAT). (2) **LogiQA** (Liu et al., 2020): LogiQA is a dataset designed to test human logical reasoning abilities, created by experts. The data is sourced from publicly available logical reasoning problems from national civil service exams. Like ReClor, each question consists of a passage, a question, and four answer choices, with only one correct option. (3) **RuleTaker** (Clark et al., 2021): RuleTaker is an automatically generated dataset of logical reasoning problems. Each problem includes a passage and a conclusion, with the task being to determine the truth of the conclusion.

Table 1: **The reasoning accuracy↑ (%) of LINA and baselines**. The Proof denotes the ProofWriter dataset. LINC is not applicable on ReClor and LogiQA, relevant results are marked as -. Bold numbers highlight the highest accuracy.

| Method | GPT-3.5-Turbo | | | | | GPT-4o | | | | |
|---|---|---|---|---|---|---|---|---|---|---|
| | ReClor↑ | LogiQA↑ | RuleTaker↑ | Proof↑ | FOLIO↑ | ReClor↑ | LogiQA↑ | RuleTaker↑ | Proof↑ | FOLIO↑ |
| Direct | 51.53 | 31.90 | 60.22 | 60.57 | 72.59 | 71.33 | 54.41 | 65.39 | 64.03 | 80.61 |
| CoT | 52.58 | 35.15 | 61.53 | 61.98 | 77.78 | 76.24 | 57.43 | 75.08 | 69.71 | 86.67 |
| CoT-SC | 55.78 | 39.22 | 64.67 | 66.59 | 79.26 | 78.19 | 58.82 | 76.47 | 74.24 | 88.11 |
| LINC | - | - | **74.25** | 59.50 | 59.12 | - | - | 82.56 | 83.64 | 78.50 |
| **LINA** | **76.6** | **51.56** | 68.01 | **71.60** | **83.46** | **86.87** | **67.96** | **89.00** | **89.41** | **93.07** |

(4) **ProofWriter** (Tafjord et al., 2021): ProofWriter is a dataset for natural language-based logical reasoning, containing 500k questions similar in style to RuleTaker. (5) **FOLIO** (Han et al., 2022): FOLIO is an expert-constructed open-domain dataset characterized by its logical complexity and diversity, used for natural language reasoning involving first-order logic. Built on the principles of first-order logic reasoning to ensure logical rigor. Each problem includes a premise and a conclusion that must be judged as true or false. we selected the validation set of ReClor (500 samples), the complete set of LogiQA, a randomly sampled subset of 1,000 examples from the validation sets of RuleTaker and ProofWriter, as well as the train set of FOLIO (1000 samples) for evaluation.

**Baselines.** We selected four prompting methods and two neuro-symbolic methods as baselines for our experiments. The prompting methods are: (1) **Direct**: Directly answering questions from the dataset using LLMs. (2) **CoT** (Kojima et al., 2022; Wei et al., 2024; Nye et al., 2021): Using chain-of-thought prompting, where LLMs generate step-by-step reasoning before answering the questions. (3) **CoT-SC** (Wang et al., 2023): Using both chain-of-thought reasoning and majority voting, where LLMs generate multiple answers, and the most frequent one is selected. (4) **ToT** (Yao et al., 2024): Transforming the LLMs' reasoning process into a search tree. The neuro-symbolic methods are: (5) **LINC** (Olausson et al., 2023): Transforming context text and conclusions into first-order logic expressions (FOLs) via LLMs, and using the FOL solver Prover9 to verify the correctness of the conclusions. (6) **SatLM** (Ye et al.): Transforming context text and conclusions into SAT code via LLMs, and using the SAT solver Z3 to verify the correctness of the conclusions. We evaluated our method against these six baselines on the five datasets mentioned above.

In principle, LINA places no restrictions on the type of LLM used. Here, we employ the most advanced GPT-4 and GPT-3.5-turbo as the base models to test the upper limits of LLM-based logical reasoning. By default, we set the temperature to 0.3 and CoT-SC to 1.0 ($n$=10).

## 5.2 REASONING PERFORMANCE EVALUATION

As shown in Table 1, our main results compare the accuracy of our method against four different baselines across five datasets. Overall, except for RuleTaker on GPT-3.5-Turbo, where its accuracy was slightly lower than that of LINC, LINA significantly outperforms all baselines. While all methods show notable improvements in accuracy over the Direct method (which uses GPT-3.5-Turbo or GPT-4 without any reasoning techniques), LINA consistently surpasses all baselines on the same model. Although LINC briefly outperforms LINA on the RuleTaker dataset using GPT-3.5-Turbo, LINA quickly surpasses LINC when tested on the same dataset using GPT-4.

It should also emphasize that on the most challenging datasets, i.e., ReClor and LogiQA, LINC's dependence of first-order logic solver prevent it from being effectively deployed. As a result, LINC's performances on these datasets are unavailable. Moreover, all baselines including CoT-SC perform poorly on the challenging LogiQA dataset, with accuracies below 40% (GPT-3.5-Turbo) and 59% (GPT-4). The complexity of LogiQA, characterized by multi-step reasoning and diverse reasoning tasks for each option, presents a significant challenge for the reasoning capabilities of the models. However, LINA addresses this by employing the hypothetical-deductive method, which processes the reasoning tasks for each option individually and leverages first-order logic rules in an LLM-driven manner throughout the reasoning process. This approach substantially improves reliability, elevating the accuracy on LogiQA to 51.56% (GPT-3.5-Turbo) and 67.96% (GPT-4).

In addition, LINA also exhibits strong performance on RuleTaker, ProofWriter, and FOLIO. In comparison, LINC's performance on these datasets is inconsistent, particularly on GPT-3.5-Turbo,


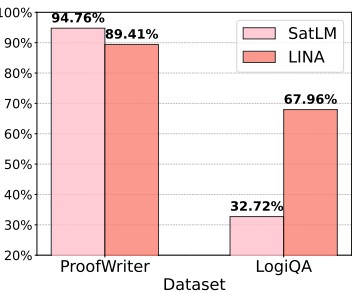
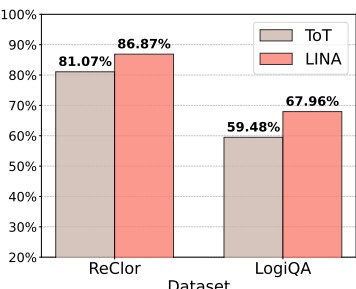

Figure 3: Comparison between LINA and SatLM on the ProofWriter and LogiQA dataset.

Figure 4: Comparison between LINA and ToT on the ReClor and LogiQA dataset.

where it underperforms the Direct method on ProofWriter and FOLIO. This phenomenon can also be attributed to LINC's too much dependence on first-order logic solver. We discover that LINC repeatedly converts the input until the solver can correctly process the derived first-order logic expressions. However, this process often results in information loss or conversion errors during the transformation phase, leading to solver failure. In contrast, LINA avoids these issues by forgoing solvers with strict input requirements and retaining the natural language information that cannot be easily transformed into first-order logic. As a result, LINA achieves more stable and improved accuracy across these datasets compared to the other baselines.

## 5.3 COMPARISON TO SATLM

As illustrated in Figure 3, we executed SatLM based on GPT-4o on the ProofWriter and LogiQA datasets. The results indicate that on the highly standardized, programmatically generated ProofWriter dataset, SatLM performs slightly better than LINA. However, on the more challenging LogiQA dataset, which involves more complex and diverse questions, LINA significantly outperforms SatLM, with the latter achieving an accuracy lower than even the Direct method, recording a score of 54.41%.

An analysis of SatLM's reasoning process reveals that its approach—using large language models to generate code for the z3 solver—can only effectively address problems where the question stem imposes strong constraints and the answer options correspond to specific constraint satisfaction scenarios. However, for more flexible question formats, such as "Which of the following can best strengthen the above argument?", the large language model is unable to generate effective z3 solver code. Consequently, SatLM fails to provide valid answers, leading to low accuracy. This issue primarily stems from the limitations in the expressiveness of the z3 solver code, which SatLM relies on. While the z3 solver is a powerful tool for solving various types of constraint-based problems, it lacks the capability to handle the flexible logical reasoning scenarios present in the LogiQA dataset, thus leading to a decline in SatLM's performance. Additionally, it should be noticed that the code generated by SatLM exhibits significant challenges in transferring across datasets. Even when applied to FOLIO, the code produced by SatLM could not be easily adapted for execution. This observation aligns with the deployment challenge mentioned in this paper, where solver-based approaches often face difficulties in deployment across different contexts.

## 5.4 COMPARASION TO TOT

As illustrated in Figure 4, we developed Tree of Thoughts (ToT) code based on the GPT-4o model to evaluate performance on the two most challenging datasets, ReClor and LogiQA, which are characterized by their complex reasoning processes. Our ToT procedure entails generating multiple distinct one-step inferences based on the information provided by the logical reasoning problem, followed by pruning performed by another large language model. This iterative process is repeated until a solution is derived.

Analysis of the experimental results reveals that, while the ToT method demonstrates superior performance compared to CoT-SC, LINAstill maintains a significant advantage over the other two meth-

Table 2: **Ablation Study Results:** The reasoning accuracy↑ (%) of LINA and ablation models on ReClor, LogiQA, RuleTaker, ProofWriter, and FOLIO. Base model is GPT-4o. Bold numbers represent the best performance in each column.

| Method | ReClor↑ | LogiQA↑ | RuleTaker↑ | ProofWriter↑ | FOLIO↑ |
|---|---|---|---|---|---|
| LINA w/o FOL | 83.43 | 62.17 | 74.80 | 81.58 | 87.12 |
| LINA w/o NL | 78.66 | 56.00 | 86.28 | 84.46 | 90.44 |
| LINA w/o Deductive | 76.38 | 43.64 | 67.35 | 64.24 | 84.43 |
| **LINA** | **86.87** | **67.96** | **89.00** | **89.41** | **93.07** |

ods in the testing datasets. This observation further corroborates the effectiveness of our approach, which integrates first-order logic expressions with natural language information and employs deductive reasoning methods, representing a notable advancement over the traditional forward reasoning processes utilized in ToT.

## 5.5 ABLATION STUDY

In addition to these baselines, we conducted an ablation study to evaluate the impact of each component of the proposed method. All ablation experiments were performed on the GPT-4o model. The ablation variants include: (1) **LINA w/o FOL**: A version of LINA without first-order logic expressions, where the logical extraction module is removed, and the model directly uses the original $Context$ and hypothesis $H$ for reasoning. (2) **LINA w/o NL**: A version of LINA without natural language information, where the extracted natural language information from the context is discarded, retaining only the logical statements $LS$, and reasoning is performed using the tuple $< LS, H >$. (3) **LINA w/o Deductive**: A version of LINA without the deductive reasoning module, where the hypothesis extraction module is removed, and forward reasoning is performed for a fixed number of $k$ steps based on the tuple $< LS, NL >$, with the intermediate reasoning process provided to the large language model as reference for answering.

The results demonstrate the importance of converting to first-order logic, retaining natural language information, and using the hypothesis-deductive reasoning strategy. The model without the logic extraction module cannot utilize first-order logic rules, leading to a lack of rigor in the reasoning process and difficulties in verifying the reasoning steps. Models without natural language information can only process information that is easily convertible into first-order logic, which leads to significant information loss, especially on challenging datasets like LogiQA. As a result, LINA w/o NL shows a performance drop of 8.21% on ReClor and 11.96% on LogiQA compared to LINA, underscoring the superiority of LINA over previous neuro-symbolic methods. However, since predicates in first-order logic expressions inherently carry some semantic information, such as in the expression `DrinkRegularly(x, coffee)`, where large language models can easily infer that $x$ regularly drinks coffee, this is difficult to avoid. This also explains why LINA w/o NL does not experience a significant performance drop on simpler datasets. Given this factor, purely neuro-symbolic approaches, which rely solely on extracting structured information, are likely to perform even worse in these cases. LINA w/o Deductive, which removes the crucial deductive reasoning module, performs poorly across multiple datasets, with performance similar to the Direct method.

## 5.6 CASE STUDY

We conduct a case study to showcase the characteristics of LINA in information extraction. As shown in Figure 5, we selected an example from the LogiQA dataset. In this example, the *context* refers to the information provided in the problem text, while the *inference* is a statement derived from one of the problem options that needs to be judged as true or false. For comparison, we selected LINC, which also utilizes first-order logic (FOL) expressions as an intermediate representation. As highlighted by cyan in the example, by only retaining the FOL expressions for the solver, LINC represents "One of them is lying" as $L(A) \lor L(B) \lor L(C) \lor L(D)$, which leads to information loss due to the lack of further elaboration on $L(x)$. In contrast, LINA preserves the natural language information "One of the statements is false" as a useful piece of information for validating the subsequent deductive reasoning process, ensuring that no information is lost at the start of the reasoning process. Moreover, LINA preserves two pivotal statements by natural language, as highlighted by

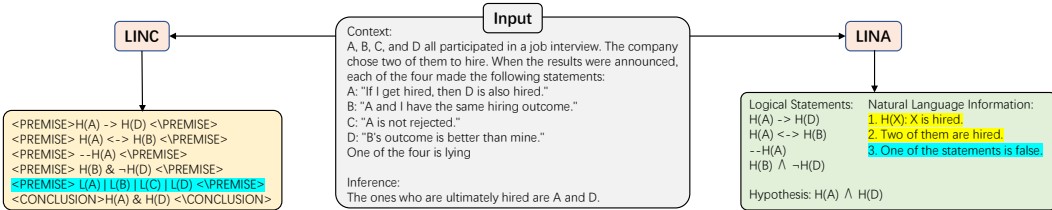

Figure 5: A comparative case of LINC and LINA from the LogiQA dataset. Text highlighted in cyan represents different content expressed by the two methods, while text highlighted in yellow represents content that is unique to one of the methods.

yellow in the example. Nevertheless, these statements are difficult to be represented by FOL. These observations demonstrate the ability of LINA to preserve valuable information for faithful logical reasoning.

## 6 CONCLUSION

In this work, we introduced LINA, an LLM-driven neuro-symbolic method for faithful logical reasoning. First, we proposed a novel information extraction approach that integrates first-order logic (FOL) expressions with natural language information. This method enables the use of FOL's rich reasoning rules during inference without sacrificing semantic content. Second, we addressed the deployment challenges of previous neuro-symbolic methods by incorporating an LLM-based reasoning module. We introduced the hypothetical-deductive method and multiple verification mechanisms to ensure the reliability of the reasoning process of LINA. Furthermore, We provide a graph interpretation of our approach and offer a detailed analysis of its properties and time complexity. The experimental results demonstrated that LINA achieves the highest accuracy across several logical reasoning benchmarks. Notably, the improvements are particularly significant on datasets with complex question structures, such as ReClor and LogiQA, further advancing the flexibility and effectiveness of LLM-driven logical reasoning.

## 7 DISCUSSION

While LINA demonstrates strong accuracy across various datasets, logical reasoning based on large language models (LLMs) remains a significant research problem. One key limitation of this work is the expressive capacity of first-order logic (FOL). Although our strategy of retaining partial natural language information prevents information loss during the extraction process, the restricted expressive power of FOL imposes constraints on the reasoning process. In problems with more complex logical structures, problem information cannot always be effectively translated into FOL, which hinders LINA from fully utilizing FOL reasoning rules to aid the inference process. Addressing this issue may require formal methods such as higher-order logics Miller & Nadathur (1986) Higginbotham (1998) or nonclassical logics Priest (2008) Burgess (2009) that can better capture the underlying logical structures of such problems. Another limitation arises from the lack of comprehensiveness in the deductive reasoning steps. Specifically, the method of selecting premises from the available conditions for each deductive step remains unresolved. This often leads to prolonged deduction processes, thereby increasing the likelihood of errors. In addition, our approach necessitates additional costs to ensure the accuracy and reliability of the reasoning process, leaving room for future enhancements in reasoning efficiency.

In future work, we will explore alternative formal methods to replace FOL, aiming to improve performance on more complex problems. We will also continue to refine the hypothetical-deductive method or investigate other systematic approaches to enhance the reliability of LLM-based reasoning. Our goal is to enable LLMs not only to retain their inherent generalizability advantages over external solvers but also to improve their accuracy in logical reasoning tasks.

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

# A    PROOF OF LEMMA AND THEOREM

## A.1    PROOF OF LEMMA 1.

First, we prove that any closed-form logical reasoning problem can be abstracted as a graph search problem: we can abstract the propositions involved in the logical reasoning problem as vertices of the graph. The proposition represented by the text of the problem statement corresponds to the initial point $s$ in the graph, and the propositions corresponding to each option correspond to the terminal points $t_{1,\ldots,n}$. Proving each option is equivalent to attempting to find a reasoning path from the initial point $s$ to one of the terminal points $t_i$. The reasoning process itself involves deriving propositions from the initial conditions (implication edges $E_t$) and using proposition equivalences (equivalence edges $E_c$) until reaching the proposition to be proven. Therefore, the reasoning process is equivalent to walking along the black edges of the graph. Thus, a closed-form logical reasoning problem has already been abstracted as the graph search problem described in Lemma 1.

Next, we prove that any graph search problem can be transformed into a closed-form logical reasoning problem: for a graph $G = (V, E_c, E_t, E_n)$, we can correspond the starting point $s$ to the problem statement in the closed-form logical reasoning problem, and the terminal point $t$ to the final conclusion of the reasoning problem. The path formed by the black edges in the graph represents the reasoning process.

This completes the proof of Lemma 1.

## A.2    PROOF OF LEMMA 2.

First, it is clear that the reasoning module of LINA is based on the assumption that both the problem statement information $< LS, NL >$ and the option hypothesis $H$ are correct. Deductive reasoning is then conducted with the goal of obtaining a reasoning result that contradicts either the problem statement information or the option hypothesis, thereby proving the falsity of the option hypothesis $H$.

The problem statement information $< LS, NL >$ corresponds to the node $q$ in the graph, and the option hypothesis $H$ corresponds to the node $a$. Since the reasoning in LINA starts by assuming both are correct, the starting point in the graph is $q \wedge a$. The search terminates at a node that contradicts the problem statement, i.e., $\neg q$, or a node that contradicts the option hypothesis, i.e., $\neg a$. This completes the proof of Lemma 2.

## A.3    PROOF OF THEOREM 1.

We use proof by contradiction. Let $p = s \wedge t$, and assume that both $s \rightarrow t$ and $p \rightarrow \neg s$ hold. Since $s \rightarrow t$ is true, we have $s \rightarrow t \wedge s$, meaning $s \rightarrow p$ is true. Furthermore, given $p \rightarrow \neg s$, by

the transitivity of logical implication, we derive $s \rightarrow s'$, which is a contradiction! Therefore, there cannot exist a path starting from $s \wedge t$ and ending at $\neg s$. The case for $\neg t$ follows similarly. This completes the proof of Theorem 1.

# B  FULL SET OF PROMPTS

## B.1  PROMPTS FOR INFORMATION EXTRACTION MODULE

---

**Prompt for Context Classification**

Please simplify the following logic problem statement and convert it into a formal logical expression. Extract the core information from each statement and present its logical structure in a concise form. Ensure that the simplified information maintains all logical relationships of the original statement, and use the following output format:

Logical Statement 1: Simplified Statement 1
Logical Statement 2: Simplified Statement 2
Logical Statement 3: Simplified Statement 3
...

You can add "Other Infomation:" items if you think there is some information that is important but is not appropriate to parallel with the Logical Statements.

---

**Prompt for FOL Translation**

**Task:** Convert the following natural language paragraph into standard first-order logic expressions. Focus on expressing the most direct and easily understandable relationships using first-order logic. Leave sentences that are difficult to express concisely in first-order logic as they are.

- Use standard logical symbols $\wedge, \vee, \rightarrow, \neg, \forall, \exists$ to represent logical relationships.
- Define and use predicates such as $P(x)$, $Q(x, y)$, etc., to represent objects or relationships.
- Appropriately use quantifiers  and  to express universal or existential statements.
- Please make sure every word in the input is showed in your output, your task is only to add some simple first-order logic expressions.

**Output Format:**

1.  Define logical predicates: Define and use predicates such as P(x), Q(x,y), etc., to represent objects or relationships.
2. Convert to first-order logic expressions: Convert only those statements that can be directly and clearly expressed in first-order logic.
3. Natural language information that is not easy to convert to FOL.

---

**Prompt for Question-Option Fusion**

Given a multiple-choice question with both a question and an option, transform them into a single proposition (a declarative statement). The proposition should combine the context provided by the question with the content of the chosen option. Use the following approach:

1.Treat the question as the background or context of the statement, removing any interrogative form or question marks.
2.Incorporate the option into the background as the key point or focus of the statement.
3.If the question involves a negation (e.g., 'except', 'false', etc.), clearly indicate that the chosen option does not satisfy the context given by the question.

---

## B.2  PROMPTS FOR LLM-DRIVEN NEURO-SYMBOLIC REASONING MODULE

---

**Prompt for Deductive Reasoner**

You are solving a logical reasoning problem that includes both a context (partly represented by first-order logic) and a proposition. Follow these steps carefully to answer the problem:

Step 1: Examine the proposition itself:
- Read the proposition carefully. If it contains a serious logical error within itself, directly judge it as false.

Step 2: Interpret the proposition using the first-order logic context.
- Break down the proposition into smaller logical components.
- Translate the proposition into first-order logic to match the context.
- If the defination in the context can't fully convert the proposition, you can skip this step.

Step 3: Apply One Step logical reasoning.
- One by One check if the components in the proposition exist in the context, examine if these cause contract first.
- Use hypothetical-deductive reasoning to check whether the proposition is consistent with all the logic statements (including the first-order logic and other natural language sentences) from the context.
- For each logical condition in the context, verify if the proposition satisfies or contradicts any condition.
- Use first-order logic rules to help you.
- You should only perform one small step of reasoning

---

**Prompt for Supervisor**

You are a supervisor tasked with overseeing the reasoning process. The goal is to evaluate whether the current Reasoning Process conflicts with the problem statement information in the Context. You will follow these steps:

1. **Check for errors**:
- If the Reasoning Process contains errors or contradictions, adjust the Reasoning Process accordingly or reset it to align with the hypothesis in the Context.

2. **Evaluate the Reasoning Process**:
- If the Reasoning Process conflicts with any part of the Context, this means the hypothesis has been refuted.
- If the Reasoning Process is fully supported by the Context, it means the hypothesis has been proven.

3. **Decision-making**:
- Based on your evaluation, decide whether to continue the reasoning process:
- If the Reasoning Process conflicts with the Context, this is a reason to stop, as the hypothesis is refuted.
- If the Reasoning Process is already supported by the Context, you may conclude the process as the hypothesis has been proven.

Continue reasoning only if the Reasoning Process neither contradicts the Context nor fully proves the hypothesis.

---

**Prompt for Reasoning Process Judgment**

You just received the text context of a logic-based multiple-choice question, the question itself, some options, and a student's analysis of these options. The student incorrectly believes that all the options are correct, but in reality, only one option is correct. Your task is to:

1. Analyze the student's reasoning for each option one by one.

2. Determine whether there are any logical errors in the student's reasoning, and point out the specific mistakes. If there are no errors, write "No mistake."

3. Finally, based on your analysis, identify the one correct option and explain why it is correct, as well as why the other options are incorrect.

Your output should follow this format:

Option 1: xxx

Error in Analysis 1: Analyze whether there is an error, pointing out specific reasons for any mistakes or stating "No mistake." Option 2: xxx

Error in Analysis 2: Analyze whether there is an error, pointing out specific reasons for any mistakes or stating "No mistake." ...

Correct Option: Write the one option you believe is correct here based on your analysis, please output the option's content, don't use the number.

