# OpenReview forum: "LINA: An LLM-driven Neuro-Symbolic Approach for Faithful Logical Reasoning"
_ICLR.cc/2025/Conference — Submitted to ICLR 2025_

### Official Review · Reviewer_FSUy · 2024-10-18

**Soundness:** 1
**Presentation:** 2
**Contribution:** 1
**Rating:** 3
**Confidence:** 4

**Summary:**

This paper proposes a pure prompt-based framework that solves reasoning problems, namely LINA. The framework first prompts LLM to convert problem into formal logic representation with natural language information; then it solves the problem as a deductive reasoning task by iteratively prompt the reasoner for deducing new facts and the supervisor for verification.

**Strengths:**

See below

**Weaknesses:**

## Novelty

The proposed method is a pure prompt-based framework with a straightforward design. The specific design of performing reasoning without using external tools has been studied in several prior works [1,2]. That said the novelty of this work is minor.


## Quality

The idea of "removing the tool usage yields better performance for deductive reasoning" is poorly motivated and justified

L48 "First, the process of converting logical problems into formal expressions leads to information loss"
- This is true for problems without FOL groundtruth, such as ReClor and LogiQA which are evaluated in the experiments.
- However, **these problems are not meant to be solved with the traditional formal logic method in the first place**, the prior work such as SatLM and LogicLM mostly focuses on solving the NLI task with datasets that come with groundtruth FOL annotations. Also note that ReClor and LogiQA contain not only deductive reasoning but also other reasoning tasks that cannot be characterized by FOL.
- That said, criticizing translation leads to information loss is fine, but it hardly motivates the approach proposed here if it is meant to solve problems that already fall outside of the formal logic bucket.

L78 "Second, the reliance on specific external tools results in poor generalization of these methods, limiting them to solving only certain types of problems, such as FOL propositional inference problems or satisfiability problems"
- This statement is problematic. Many works show tool usage increases rather than decreases the capability of LLMs in solving formal reasoning problems.
- Formal tools such as Prover9 and Z3 can be used for not only propositional logic but also first-order logic. And sat problem is a very generic problem setting where many reasoning problems can be converted into a sat problem, and being able to solve sat problem should be not considered as a disadvantage.
- That said, the authors should motivate their work properly.


Not every reasoning problem in ReClor and LogiQA can be formed into deductive reasoning:
- The authors propose to solve all reasoning problems with deductive reasoning. This is simply inappropriate for many of the problems in ReClor and LogiQA. For example, ReClor contains questions like "which of the following most challenges/supports/aligns with the argument in the context?" and "which of the following arguments shares the same reasoning pattern as that in the context", such questions do not fit into any formal logic categories and certainly cannot be solved with deductive reasoning.


The experiment setting misses many details and is potentially problematic:
- It's unclear how many ICL examples are used for GPT CoT baselines. However, an accuracy of 76 with GPT-4o on ReClor seems too bad to be true. As a comparison, [3] shows that with just a few ICL examples, GPT-3.5 can achieve about 60% accuracy and GPT-4 can achieve above 90% accuracy, which aligns much better with the scores reported in the public leaderboard.
- As mentioned above, including methods like LINC in ReClor and LogiQA benchmarks is not sensible, as these methods are designed for NLI task and not these benchmarks.



## Clarity

The paper is generally easy to follow.


## Significance

While I agree with the authors that moving beyond standard NLI tasks into more "in the wild" reasoning problems such as that in ReClor is an interesting and important direction, it cannot justify the pure prompt-based design, as it effectively rendering the approach into yet another fancy CoT method that could hallucinate during its reasoning. From a pure performance perspective, the significance of this work is still questionable as the results from the baseline approach are too bad to be true. That said, the significance is also minor.


[1] Zhu, Zhaocheng, et al. "Large language models can learn rules." arXiv preprint arXiv:2310.07064 (2023).

[2] Feng, Jiazhan, et al. "Language models can be logical solvers." arXiv preprint arXiv:2311.06158 (2023).

[3] Yang, Yuan, et al. "Can LLMs Reason in the Wild with Programs?." arXiv preprint arXiv:2406.13764 (2024).

**Questions:**

see above

---

> ### Author Response · Authors · 2024-11-25
>
> We thank you very much for your detailed and constructive review. We hope the following response could address your concerns.
>
> **Q1: Could the authors explain the novelty of this paper compared to [1] and [2]?**
> - **[1]:** This work employs fine-tuning to enable reasoning in natural language and aims to make LLMs function like symbolic solvers. In contrast, LINA’s Information Extraction module retains both FOL expressions and natural language information, offering the advantages of symbolic methods without requiring fine-tuning.
> - **[2]:** This approach teaches LLMs reasoning rules before reasoning. LINA focuses on a novel information extraction method, leveraging symbolic information to assist LLM reasoning. Additionally, the introduction of the hypothetical-deductive method enhances generalizability and performance, marking a key contribution of our work.
>
>
>
> **Q2: Could the authors explain their research motivation?**
> We acknowledge the significant contributions of works like LINC and SatLM, which integrate external solvers into symbolic methods. However, these works primarily utilize LLMs for language understanding, converting logical reasoning tasks into formats comprehensible to specialized solvers.
>
> LINA aims to exploit LLMs' strong generalization and reasoning capabilities, addressing limitations where solvers are task-specific. We propose innovative information extraction and reasoning methods, reinforcing LLMs’ reasoning ability and improving performance.
>
>
> **Q3: Why did the authors choose deductive reasoning to address problems in ReClor and LogiQA that don't belong to formal logical categories?**
> Our approach goes beyond simple deductive reasoning by employing the **hypothetical-deductive method**, a general-purpose reasoning strategy. In closed-choice questions, we treat each Option as a hypothesis, integrating it with Question context for evaluation. For example, when addressing "Which option most challenges the argument in the Context?", we extract each Option’s perspective, assume it to be the most challenging viewpoint, and use the Reasoning module to test its validity.
>
> This flexible approach is effective even when formal logical categories are not strictly applicable.
>
>
> **Q4: Could the authors explain the complexity of their method, given that such designs might lead to hallucinations?**
> Our modular reasoning process is explicitly designed to **reduce hallucinations**, not increase them.
> 1. **Deductive Reasoner:** Employs step-by-step reasoning combined with a Supervisor to verify FOL-based reasoning rules, mitigating biases seen in multi-step CoT reasoning.
> 2. **Reasoning Process Judgment:** Provides an additional layer of verification, refining outputs to ensure only the most reliable conclusions are presented.
>
> Thus, the complexity reflects our effort to minimize hallucinations and enhance reasoning reliability.
>
> If any concerns remain unresolved, please feel free to let us know. Thank you again for your time and patience.

---

> > ### Comment · Reviewer_FSUy · 2024-12-02
> > **Response**
> >
> > Thanks for the response. Unfortunately, I think this concept of *hypothetical-deductive method* is not a well-defined concept, nor is it a well-justified approach to these problems. What's the benefit of it? Why is it a valid method for the ReClor-type problems? This is implicitly assumed in the paper and is only brought up in the rebuttal. To better motivate this, one needs to rewrite the paper to center around this concept and carefully design the experiments and ablations for justification, and this requires an overhaul of the draft and another round of reviews to make sure things are in place. That said, I keep my score.

---

### Official Review · Reviewer_7jFX · 2024-10-30

**Soundness:** 2
**Presentation:** 3
**Contribution:** 2
**Rating:** 3
**Confidence:** 4

**Summary:**

The paper proposes a framework called LINA to address the generalization problem and information loss found in existing methods. The framework consists of two main components: an Information Extraction Module and a Symbolic Reasoning Module. First, the Information Extraction Module condenses and translates the reasoning question into a symbolic format. Then, the Symbolic Reasoning Module iteratively performs one-step deductive reasoning, utilizing both symbolic and natural language, with a judgment step to verify the correctness of each reasoning step. By leveraging GPT-3.5 and GPT-4o, the paper demonstrates that LINA outperforms the baselines across five datasets. Additionally, the paper includes comparisons to ToT and SatLM, along with an ablation study and a case study.

**Strengths:**

(1)	The framework claims to effectively address the issue of poor generalization to different question formats and the problem of information loss when using only symbolic language by combining symbolic and natural language.

(2)	The main experiment shows that the method surpasses the baselines across five datasets using GPT-3.5 and GPT-4o.

**Weaknesses:**

(1)	The level of innovation in this work raises some concerns. To the best of my knowledge, some previous work (SymbCoT [3]), also addresses the issue of information loss by leveraging both natural language information and first-order logic (FOL). The key difference is that SymbCoT conducts reasoning and verification as a linear process, whereas this work transforms the process into an iterative one. In summary, it appears that this work mainly modifies the linear process from SymbCoT into an iterative framework, which limits its novelty. From my perspective, the primary innovation lies in this framework’s adaptability to a wide range of question formats, a feature the previous work lacks.

(2)	The Information Extraction Module requires further clarification. How is the context classified? Additionally, how do you determine the "ease of translation"? Upon reviewing the context classification prompt provided in the appendix, it seems more focused on simplifying logical statements rather than classification. Please clarify if my understanding is incorrect.

(3)	In Section 4.2, you explain that the context is first classified into lengthy text and non-lengthy text, with the lengthy text then being condensed into shorter sentences. These condensed texts are further classified based on their ease of translation. Further details are needed to understand this process. For example, how many classes are used in this step? Which classes will be translated, and which will not? This is important because the paper claims an advantage in using both symbolic and natural language, so it is crucial to understand what content is represented in symbolic language and what remains in natural language.

(4)	The Reasoning Module lacks crucial details. Firstly, there is no explanation of how the deductive process works and how information LS, NL, and H interact to reach the reasoning conclusion C.  Secondly, when performing the Check() operation, is it checking for errors in the reasoning process, or is it verifying whether the information contradicts or supports the hypothesis? Third, you mention that if an error occurs, the supervisor may adjust C or reset C = H. How is this step implemented exactly? This is not explained in the main text nor in Algorithm 1, and more details are needed to help readers understand how the reasoning module operates.

(5)	The paper lacks a detailed analysis, which hinders the reader's understanding and the transparency of the framework. For example, the paper's main claim is that it addresses information loss and improves the framework's generalizability, but there is a lack of relevant analysis to support this claim. Besides, prior work in this stream (e.g., LINC [1], Logic-LM [2], SymbCoT [3]) typically includes an analysis of accuracy per necessary proof depth in ProofWriter. Including this type of analysis would be valuable, as it could demonstrate how robust your method is with respect to increasing reasoning complexity, a common challenge in real-world applications. Furthermore, the paper lacks an error analysis, which would provide a clearer understanding of where failures occur and improve confidence in the proposed framework.

(6)	The analysis section also lacks some details. In Section 5.4, when you state that the LLM cannot generate effective z3 solver code or easily adapt for execution, does this mean that the rule-based solver completely fails to execute the problem, or can it execute but fail to reach the correct answer? Do you have quantitative data, such as execution rates, to back up this observation?

Reference:
[1] LINC: A Neurosymbolic Approach for Logical Reasoning by Combining Language Models with First-Order Logic Provers (Olausson et al., EMNLP 2023)
[2] Logic-LM: Empowering Large Language Models with Symbolic Solvers for Faithful Logical Reasoning (Pan et al., EMNLP 2023)
[3] Faithful Logical Reasoning via Symbolic Chain-of-Thought. (Xu et al., ACL 2024)

**Questions:**

(1) Why did you choose to use the train set of FOLIO while using the validation set for other datasets? Is there a specific reason for this decision? Most prior work (e.g., Logic-LM) typically evaluates the test set of FOLIO, so it would be helpful to clarify the rationale behind this choice.

(2) Could you provide more details about how the hypothesis is generated? Additionally, could you elaborate on how the Reasoning Process Judgment is integrated into the framework? It appears in Figure 1 but is not included in Algorithm 1, which causes some confusion. Providing more information on this would make the methodology easier to follow for readers.

(3) Do you have quantitative data to support your claim?

---

> ### Author Response · Authors · 2024-11-25
>
> We thank you very much for your detailed and constructive review. We hope the following response could address your concerns.
>
> **Q1: How are the contribution and novelty of this work?**
> Our method differs from SymbCoT in its information retention mechanism. While SymbCoT iteratively conducts symbolic reasoning, our approach emphasizes the hypothetical-deductive process.
>
> Our Information Extraction module preserves both first-order logic (FOL) expressions and critical natural language information, allowing our method to utilize symbolic information effectively rather than merely treating the LLM as a symbolic reasoner.
>
> Additionally, our reasoning framework is not a simple "Plan-and-Solve" structure. Recognizing the limitations of purely symbolic reasoning with LLMs, we have designed a framework based on the scientific method of hypothesis-deduction, enhancing the reliability of LLM reasoning. Therefore, our method is not a simple iterative symbolic reasoning approach.
>
> **Q2: Is the Information Extraction module more focused on simplifying logical statements rather than classification? Explain the module in detail.**
> The Information Extraction module operates in three steps:
> 1. Simplifying all text to enhance clarity.
> 2. Classifying text based on its ease of transformation into FOL expressions.
> 3. Transforming the easily convertible parts into FOL expressions while retaining critical natural language information.
>
> This approach ensures effective extraction of logical and textual information.
>
> **Q3: Why does FOLIO use the training set instead of the validation set?**
> The FOLIO training set contains 1000 samples, while the validation set only has 203. The larger size of the training set aligns better with the scale of other datasets, allowing for a more comprehensive evaluation of the method's performance.
>
>
> **Q4: Could the author provide more details on hypothesis generation, the final module integration, and their relationship with the algorithm?**
> Certainly.
> - **Hypothesis Generation:** This process integrates information from the Question and Options, reformulating them into declarative propositions. Closed-choice questions often embed critical information in the Question, necessitating integration with the Options to form hypotheses.
> - **Reasoning Process Judgment:** This module operates after the Deductive Reasoner. Since each closed-choice question generates multiple hypotheses, errors in reasoning may lead to multiple plausible answers. This module evaluates the outputs of the Deductive Reasoner to select the final correct option.
>
>
> **Q5: Could the author provide finer-grained quantitative data, include experiments on PW depth, conduct error analysis, and analyze cases where the Z3 solver fails?**
> Thank you for the suggestion. We will include these additional experiments and analyses in future work.
>
> If any concerns remain unresolved, please feel free to let us know. Thank you again for your time and patience.

---

> > ### Comment · Reviewer_7jFX · 2024-12-02
> >
> > Thank you for your response. The central claim of your paper, which proposes combining FOL and natural language to address information loss, has already been explored in existing research. As a result, the technical contribution of the work appears to be limited.
> >
> > I agree with Reviewer FSUy’s point that, if the primary distinction of your paper lies in the hypothesis-deduction framework, the paper should be restructured to emphasize this as the core contribution. Additionally, the current experimental design does not robustly support this claim. To strengthen the paper, I recommend redesigning the experiments to more clearly demonstrate the value of the "hypothesis-deduction" framework.
> >
> > Given these points, I will maintain my current rating.

---

### Official Review · Reviewer_itSY · 2024-11-02

**Soundness:** 2
**Presentation:** 3
**Contribution:** 2
**Rating:** 5
**Confidence:** 4

**Summary:**

The paper introduces LINA, a neuro-symbolic approach designed to enhance the logical reasoning abilities of LLMs. LINA implements a hypothetical-deductive reasoning paradigm by enabling LLMs to autonomously manage logical reasoning without external solvers. It extracts propositional logic from natural language, and performs deductive logical reasoning. Empirical results show LINA outperforms existing methods, including LINC and other prompting techniques.

**Strengths:**

Improving LLM-based reasoning with neuro-symbolic integration is a good research problem. The writing is well-structured and clear. Empirical results are given with details. Code and data are provided for reproducibility.

**Weaknesses:**

The core concept of the proposed approach is an agentic framework equipped with formal logic, which is relatively common. The advantages of translating natural language into formal logic and using LLMs for reasoning remain ambiguous. The effectiveness of the agentic framework is influenced by the capabilities of base model and potential self-bias. The application scope of the method is limited.

**Questions:**

1) The authors propose an agentic framework that utilizes formal logic to enhance LLMs. Could a broader comparison with other relevant approaches (in addition to LINC) [1-4] be considered to provide a more comprehensive evaluation?


2) LLMs are generally stronger in processing natural language compared to formal logic. Could the authors clarify the advantages they see in converting logical reasoning tasks from natural language into Propositional or First-order Logic for LLM-based reasoning? If this conversion strategy offers benefits, might it be more effective to prompt LLMs with Chain-of-Thought reasoning including Propositional or First-order Logic?


3) The authors introduce an agentic framework for symbolic reasoning without an external solver. Could they explain the rationale behind this choice in more detail? If the concern is that formal logic generated by LLMs may be unreliable for external solvers, how does the proposed framework address this issue? Additionally, since the agentic approach relies on a sufficiently capable base model for sub-task management, would this framework extend well to smaller models (such as 7-8B parameters)?


4) Given that LLMs can struggle with self-bias [5], could the authors discuss any potential limitations in having the same LLM serve as both the deductive reasoner and supervisor/judge? Are there mechanisms in place to help mitigate self-bias and enhance the model's verification process?


5) One challenge with deduction using formal logic can be the restricted scope, especially if the required deduction rules are not explicitly included as known information. Could the authors share any strategies to address this challenge? Additionally, do they see any potential for extending this formal logic framework to reasoning tasks that require broader expressiveness, such as math reasoning, coding, and question answering?


Reference:

[1] Pan, Liangming, et al. "Logic-LM: Empowering Large Language Models with Symbolic Solvers for Faithful Logical Reasoning." The 2023 Conference on Empirical Methods in Natural Language Processing.

[2] Yang, Sen, et al. "Neuro-symbolic integration brings causal and reliable reasoning proofs." arXiv preprint arXiv:2311.09802 (2023).

[3] Xu, Fangzhi, et al. "Symbol-LLM: Towards foundational symbol-centric interface for large language models." arXiv preprint arXiv:2311.09278 (2023).

[4] Xu, Jundong, et al. "Faithful Logical Reasoning via Symbolic Chain-of-Thought." arXiv preprint arXiv:2405.18357 (2024).

[5] Huang, Jie, et al. "Large Language Models Cannot Self-Correct Reasoning Yet." The Twelfth International Conference on Learning Representations, 2024.

---

> ### Author Response · Authors · 2024-11-25
>
> We thank you very much for your detailed and constructive review. We hope the following response could address your concerns.
>
> **Q1. Could the authors include a broader comparison with approaches beyond LINC (e.g., [1-4]) for a more comprehensive evaluation?**
> Thank you for your valuable suggestion. We have already compared LINA with LINC, SatLM, and various prompting-based methods. We will expand our comparison in the revised version. One notable advantage of LINA over [1-4] is its hybrid approach, which combines symbolic reasoning with natural language representations. This design preserves semantic richness while leveraging the benefits of formal logic.
>
> **Q2. Could the authors clarify the benefits of converting logical reasoning tasks into Propositional or First-Order Logic (FOL) for LLM reasoning? Would incorporating such logic into Chain-of-Thought (CoT) prompting be more effective?**
> Certainly. The primary advantage lies in separating reasoning from natural language, allowing LLMs to operate in the formal logic space using well-defined rules. This process ensures more reliable reasoning and facilitates error detection.
> As shown in our ablation studies, symbolic representations combined with CoT already improve LLM performance. However, our proposed hypothesis-deduction mechanism further enhances reasoning ability, and its contribution should not be overlooked.
>
> **Q3. Why did the authors choose a symbolic reasoning framework without an external solver? How does the framework handle unreliable formal logic from LLMs, and can it work with smaller models (e.g., 7-8B parameters)?**
> We appreciate this insightful question. Our primary focus was addressing two issues related to external tools: **information loss** (L49) and **poor generalization** (L81). External solvers often require strict input formats, which may not fully capture the original semantics of natural language. By retaining critical natural language information alongside FOL expressions, our approach mitigates this problem while leveraging LLMs’ rule-based and semantic reasoning abilities.
> To handle unreliable formal logic generation, we designed an information extraction module that reduces errors by preserving natural language alongside FOL. This strategy ensures that translation inaccuracies do not entirely compromise reasoning.
> Regarding smaller models, we plan to conduct additional experiments to explore the framework's adaptability.
>
> **Q4. Since LLMs may struggle with self-bias, could the authors address limitations of using the same LLM as both reasoner and judge?**
> This is a valid concern. To mitigate self-bias, we introduced FOL representations to transform error-prone natural language reasoning into verifiable formal logic. Additionally, we incorporated a separate Reasoning Process Judgment module, built with a different LLM, to verify the correctness of the reasoning process.
>
> **Q5. Could the authors share strategies to address missing rules in formal logic deduction and discuss extending the framework to tasks like math reasoning, coding, or question answering?**
> As shown in our experiments, explicitly instructing LLMs to use FOL rules can yield promising results. This success may stem from the relative simplicity of FOL rules, which LLMs are already familiar with.
> Regarding potential extensions, our framework’s hypothesis-deduction mechanism could adapt to other domains with structured reasoning tasks, such as math reasoning or coding. By integrating domain-specific reasoning rules, the framework could effectively tackle broader tasks.
>
> If any concerns remain unresolved, please feel free to let us know. Thank you again for your time and patience.

---

> > ### Comment · Reviewer_itSY · 2024-12-03
> > **Official Comment by Reviewer itSY**
> >
> > Thank you for your response. I believe that additional experiments for comparison would strengthen the argument. Experimental results and analysis would be more compelling than plain explanations.
> >
> > The introduction of external solvers is intended to provide a more reliable, albeit narrower, mechanism to support LLMs. From my understanding, if the rule application/program execution is simulated using prompting, we actually lose this benefit and reduce to another type of CoT reasoning, which is supposed to be worse due to insufficient training data.
> >
> > Therefore, I maintain the score.

---

### Official Review · Reviewer_7yeu · 2024-11-04

**Soundness:** 4
**Presentation:** 3
**Contribution:** 3
**Rating:** 6
**Confidence:** 3

**Summary:**

The authors propose LINA, a framework that decomposes the reasoning steps for complex questions using four main components: (1) an LLM-based logic extractor, (2) an LLM-based query extractor, (3) an LLM-powered logic deducer, and (4) a core algorithm that integrates context and derived results to analyze the correctness of the underlying answers. They also provide theoretical analysis of LINA’s properties and complexity. Experimental results demonstrate that LINA significantly improves performance on benchmarks requiring multi-step reasoning, outperforming existing methods like Chain-of-Thought (CoT) by a substantial margin.

**Strengths:**

**Originality: 4/5**

A closely related work, SatLM, uses the Z3 solver as its logical reasoning backbone, whereas LINA leverages an LLM-prompt-based approach. While both frameworks share a similar conceptual foundation, LINA’s LLM-based reasoning backbone is more adaptable to loosely defined questions, enabling it to outperform the more rigid solver approach. This novel application of an LLM-driven deductive logic engine enhances generalizability.

**Quality: 3.5/5**

Pros: The authors provide both theoretical proofs on complexity and robust experimental results across multiple benchmarks. One question that arises is how well the LLM-powered deductive logic engine performs on standard logical deduction problems.

Cons: The reported accuracy for ReClorTeam (GPT-4-0613) on the ReClor leaderboard is 90.10, which is notably different from the numbers presented in this paper.

**Clarity: 3.5/5**

Pros: Figure 1 effectively clarifies the pipeline, and the appendix, which includes the actual prompts, further aids understanding.

Cons: Figure 2 is challenging to interpret without sufficient context, and it’s unclear why the Chain-of-Thought (CoT) approach does not explore additional steps.

**Significance**

This work is of the interest for both neural symbolic community and NLP community.

**Weaknesses:**

As shown in strength.

**Questions:**

1. How well the LLM-powered deductive logic engine performs on standard logical deduction problems.
2. The reported accuracy for ReClorTeam (GPT-4-0613) on the ReClor leaderboard is 90.10, which is notably different from the numbers presented in this paper. What may cause the difference?

---

> ### Author Response · Authors · 2024-11-25
>
> We thank you very much for your constructive and insightful review. We hope the following response could address your concerns.
>
> **Q1. How well does the LLM-powered deductive logic engine perform on standard logical deduction problems?**
> Thank you for raising this question. As highlighted in our paper, we evaluated the engine on the RuleTaker dataset, a dataset of standard logical deduction problems generated using strict computer programs and logical specifications. Our approach demonstrates superior performance compared to the baselines.
>
> **Q2. Why does the reported accuracy for GPT-4-0613 on the ReClor leaderboard differ from the numbers in this paper?**
> Please refer to the Common section above for a detailed explanation.
>
> If any concerns remain unresolved, please feel free to let us know. Thank you again for your time and patience.

---

> > ### Comment · Reviewer_7yeu · 2024-11-28
> >
> > The authors' rebuttal has addressed most of my concerns. I would like to maintain the rating.

---

### Author Response · Authors · 2024-11-25

### **Common Question**
**Why does the ReClor Team and some other papers report an accuracy of 90% for GPT-4 on ReClor, while the results in our paper show 71.33% for GPT-4o?**
We appreciate the reviewer’s attention to this point. Our experimental setup for the ReClor dataset differs from that of the ReClor leaderboard. Specifically, our approach does not directly answer the multiple-choice questions. Instead, we convert the four options into four separate hypotheses and evaluate their correctness individually. Consequently, we applied the same processing methodology to the ReClor and LogiQA datasets for both Direct and CoT methods, where each option is judged separately. In cases where multiple options are deemed correct, an additional LLM round is used to select the final answer from these correct options.

This methodology aligns with work that aims to mitigate data contamination in LLMs (e.g., PertEval[1]). We believe the observed accuracy drop partly results from reducing data contamination from the LLM’s training process. To verify this, we conducted a new set of Standard experiments using GPT-4o on the ReClor dataset under our experimental setup. In this setup, each of the four options was evaluated for correctness, and instances of multiple correct options were also recorded. The results are summarized in the table below:

| **Number of Correct Options** | **Proportion** |
|-------------------------------|----------------|
| Single Option                | 51.0%          |
| Two Options                  | 25.2%          |
| Three Options                | 6.8%           |
| Four Options                 | 1.6%           |
| **Total**                    | **84.6%**      |

From the 71.33% accuracy reported in our paper, we can see that GPT-4o successfully identifies all questions where only the correct option is considered valid. It also performs well on cases with two correct options but struggles with those involving three or four correct options. Even in this challenging setup, the overall accuracy for questions with at least one correct option identified reaches 84.6%. This indicates that some level of data contamination likely exists in the ReClor dataset for GPT-4o.

Therefore, we consider our experimental design, which mitigates data contamination, to be a fair and reasonable approach.

[1] Li J, Hu R, Huang K, et al. "PertEval: Unveiling Real Knowledge Capacity of LLMs with Knowledge-Invariant Perturbations. "The 38th Annual Conference on Neural Information Processing Systems。

---

### Meta-Review · Area_Chair_Ju5A · 2024-12-18

**Metareview:**

The reviewers all felt that the paper had a lot of positives but the paper in its current form had a lot of issues as well. So the general consensus was that the paper requires another round of rewrite. If this was a journal, this paper would be categorized as major revisions but in the conference cycle, this has to be reviewed completely again.

**Additional Comments On Reviewer Discussion:**

The reviewers acknowledged that the authors gave good explanations for many of the points raised but they felt that the core issue of writing needs to be addressed.

---

### Decision · Program_Chairs · 2025-01-22

Reject